# Thalidomide promotes degradation of SALL4, a transcription factor implicated in Duane Radial Ray syndrome

Katherine A Donovan[1,2], Jian An[1,2], Radosław P Nowak[1,2], Jingting C Yuan[1], Emma C Fink[3,4], Bethany C Berry[1], Benjamin L Ebert[3,4], Eric S Fischer[1,2]*

[1]Department of Cancer Biology, Dana-Farber Cancer Institute, Boston, United States; [2]Department of Biological Chemistry and Molecular Pharmacology, Harvard Medical School, Boston, United States; [3]Division of Hematology, Brigham and Women's Hospital, Boston, United States; [4]Department of Medical Oncology, Dana-Farber Cancer Institute, Boston, United States

**Abstract** In historical attempts to treat morning sickness, use of the drug thalidomide led to the birth of thousands of children with severe birth defects. Despite their teratogenicity, thalidomide and related IMiD drugs are now a mainstay of cancer treatment; however, the molecular basis underlying the pleiotropic biology and characteristic birth defects remains unknown. Here we show that IMiDs disrupt a broad transcriptional network through induced degradation of several $C_2H_2$ zinc finger transcription factors, including SALL4, a member of the *spalt*-like family of developmental transcription factors. Strikingly, heterozygous loss of function mutations in *SALL4* result in a human developmental condition that phenocopies thalidomide-induced birth defects such as absence of thumbs, phocomelia, defects in ear and eye development, and congenital heart disease. We find that thalidomide induces degradation of SALL4 exclusively in humans, primates, and rabbits, but not in rodents or fish, providing a mechanistic link for the species-specific pathogenesis of thalidomide syndrome.

DOI: https://doi.org/10.7554/eLife.38430.001

*For correspondence:
eric_fischer@dfci.harvard.edu

## Introduction

Thalidomide was first marketed in the 1950s as a nonaddictive, nonbarbiturate sedative with anti-emetic properties, and was widely used to treat morning sickness in pregnant women. Soon after its inception, reports of severe birth defects appeared, but it was denied that these were linked to thalidomide. In 1961, two independent reports confirmed that thalidomide was causative to this, the largest preventable medical disaster in modern history (*Lenz, 1962*; *McBride, 1961*). In addition to thousands of children born with severe birth defects, there were reports of increased miscarriage rates during this period (*Lenz, 1988*). Despite this tragedy, thalidomide, and its close derivatives, lenalidomide and pomalidomide, known as immunomodulatory drugs (IMiDs), continue to be used to treat a variety of clinical conditions such as multiple myeloma (MM) and 5q-deletion associated myelodysplastic syndrome (del(5q)-MDS) (*D'Amato et al., 1994*; *Pan and Lentzsch, 2012*).

Although a potentially transformative treatment for MM, the molecular mechanisms of thalidomide teratogenicity, and many of its biological activities remain elusive. It was only recently shown that thalidomide and analogs exert their therapeutic effect by binding to the Cullin RING E3 ubiquitin ligase CUL4-RBX1-DDB1-CRBN (CRL4^CRBN) (*Chamberlain et al., 2014*; *Fischer et al., 2014*; *Ito et al., 2010*) and promoting ubiquitination and degradation of key efficacy targets (*neo*-substrates), such as the zinc finger (ZnF) transcription factors IKAROS (IKZF1), AIOLOS (IKZF3), and ZFP91 (*An et al., 2017*; *Fischer et al., 2014*; *Gandhi et al., 2014b*; *Krönke et al., 2014*; *Lu et al.,*

**eLife digest** Thalidomide was sold in the 1950s and 1960s as a sedative and anti-nausea medication for pregnant women suffering from morning sickness. Studies in mice and other animals had suggested thalidomide was safe and led some countries to allow the drug to be used in humans. By 1961, it became clear that thalidomide use by pregnant women led to serious birth defects, and the drug was removed from the market. By then, thalidomide had caused birth defects in over 10,000 babies, a tragedy that has been described as the biggest man-made medical disaster in human history. It led many countries to adopt tougher standards for drug safety.

Thalidomide and similar drugs are now used with great success to treat leprosy and various blood cancers. But questions remain about exactly how the drugs work and how they cause birth defects like shortened arms and legs. Previous studies have shown that thalidomide binds to a protein called cereblon, which marks other proteins for destruction and removal from the cell. Thalidomide hijacks cereblon and causes it to tag the wrong proteins.

To learn more about how thalidomide causes birth defects, Donovan et al. treated human embryonic stem cells and cancer cells with thalidomide and related drugs. Analyzing the proteins inside the cells revealed that the drugs caused dramatic reductions in the amount of a protein called SALL4, which is essential for limb development. It was already known that mutations in the gene that produces SALL4 cause two conditions called Duane Radial Ray syndrome and Holt-Oram syndrome. Both conditions can result in birth defects like those seen in babies exposed to thalidomide.

As well as showing that thalidomide-hijacked cereblon marks SALL4 for destruction, Donovan et al. also reveal why mice do not develop birth defects when exposed to thalidomide. This is because genetic differences make the mouse cereblon proteins unable to tag SALL4. Researchers could now build on these results to develop safer versions of thalidomide that do not target SALL4 while still successfully treating leprosy and cancers.

DOI: https://doi.org/10.7554/eLife.38430.002

---

*2014*). IMiDs can also promote degradation of targets that lack a zinc finger domain, including Casein Kinase 1 alpha (CSNK1A1) (*Krönke et al., 2015*; *Petzold et al., 2016*) and GSPT1 (*Matyskiela et al., 2016*). CRL4$^{CRBN}$ has further been implicated in the IMiD-independent turnover of GLUL, BSG, and MEIS2 (*Eichner et al., 2016*; *Krönke et al., 2014*; *Nguyen et al., 2016*; *Fischer et al., 2014*), and regulation of AMPK (*Lee et al., 2013*), processes potentially inhibited by IMiDs. Although no obvious sequence homology exists between the known IMiD-dependent CRL4$^{CRBN}$ substrates, all share the characteristic β-hairpin loop structure observed in X-ray crystal structures of IMiDs bound to CRBN and CSNK1A1 or GSPT1 (*Matyskiela et al., 2016*; *Petzold et al., 2016*), and a key glycine residue that engages the phthalimide moiety of IMiDs (*An et al., 2017*; *Matyskiela et al., 2016*; *Petzold et al., 2016*). Despite progress in understanding the therapeutic mechanism of action of thalidomide, the cause of thalidomide syndrome has remained unknown since its description in 1961. Over the last 60 years, multiple theories such as anti-angiogenic properties or the formation of reactive oxygen species (ROS) by thalidomide, or specific metabolites of thalidomide have been linked to thalidomide-induced defects. However, rarely do they explain the full spectrum of birth defects caused by all members of the IMiD family of drugs (*Vargesson, 2015*). Moreover, it has been shown that species such as mice, rats, and bush babies are resistant to thalidomide-induced teratogenicity (*Butler, 1977*; *Heger et al., 1988*; *Ingalls et al., 1964*; *Vickers, 1967*), which suggests an underlying genetic difference between species, more likely to be present in a specific substrate rather than in a general physiological mechanism such as anti-angiogenic effects or ROS production. To date, IMiD target identification efforts have largely focused on elucidating the mechanism of therapeutic efficacy of these drugs in MM and del(5q)-MDS (*Gandhi et al., 2014a*; *Krönke et al., 2015*, *2014*; *Lu et al., 2014*). However, these hematopoietic lineages may not express the specific proteins that are important in the developmental events disrupted by thalidomide during embryogenesis. In the absence of tractable animal models that closely resemble the human disease, we focused on human embryonic stem cells (hESC) as a model system that more likely expresses proteins relevant to embryo development, and set out to investigate the effects of thalidomide in this developmental context.

## Results

### IMiDs induce CRL4$^{CRBN}$-dependent degradation of multiple C$_2$H$_2$ zinc finger transcription factors

We established a mass spectrometry-based workflow (*Figure 1—figure supplement 1A*) to detect IMiD-induced protein degradation in hESC. To identify targets of IMiDs, we treated cells with 10 μM thalidomide, 5 μM lenalidomide, 1 μM of pomalidomide, or a DMSO control (*Figure 1—figure supplement 1B*). To minimize transcriptional changes and other secondary effects that often result from extended drug exposure (*An et al., 2017*), cells were treated for 5 h and protein abundance was measured in multiplexed mass spectrometry-based proteomics using tandem mass tag (TMT) isobaric labels (*McAlister et al., 2014*) (*Figure 1—figure supplement 1* and Materials and methods). From ~10,000 proteins quantified in H9 hESC, only the developmental *spalt*-like transcription factor SALL4 showed statistically significant downregulation across all three drug treatments with a change in protein abundance greater than 1.5-fold, and a p value < 0.001 (*Figure 1A–C*). In accordance with previous findings, we also observed that treatment with lenalidomide led to degradation of CSNK1A1 (*Krönke et al., 2015*; *Petzold et al., 2016*). Pomalidomide induced degradation of additional targets, including the previously characterized zinc finger protein ZFP91 (*An et al., 2017*) and the largely uncharacterized proteins ZBTB39, FAM83F, WIZ, RAB28, and DTWD1 (*Figure 1A–C*).

This diverse set of *neo*-substrates observed in response to treatment with different IMiDs (number of substrates identified: Thal < Len << Pom) prompted us to further expand our exploration of IMiD-dependent *neo*-substrates by profiling IMiDs in additional cell lines. As degradation is mediated through CRL4$^{CRBN}$, and because CRBN expression levels are high in the central nervous system (CNS), we assessed the effects of IMiDs in two different neuroblastoma cell lines, Kelly and SK-N-DZ cells, as well as the commonly used multiple myeloma cell line, MM1s, as a control. Comprehensive proteomics studies across multiple independent replicates of hESC, Kelly, SK-N-DZ, and MM1s cells (*Figure 1A–D*, see Materials and methods and *Figure 1—figure supplements 1* and *2* for details), revealed multiple novel substrates for IMiDs (ZNF692, SALL4, RNF166, FAM83F, ZNF827, RAB28, ZBTB39, ZNF653, DTWD1, ZNF98, and GZF1). To validate these novel targets, we carried out a 'rescue' proteomics experiment, in which we treated SK-N-DZ cells with 1 μM pomalidomide or with a co-treatment of 1 μM pomalidomide and 5 μM MLN4924 (a specific inhibitor of the NAE1/UBA3 Nedd8 activating enzyme). Inhibition of the Cullin RING ligase (CRL) by MLN4924 fully abrogated IMiD-induced degradation of targets (*Figure 1—figure supplement 2B,C*), and thereby confirmed the CRL-dependent mechanism. This approach was confirmed by spot-checking IMiD-dependent degradation for novel targets for which antibodies were available by western blot (*Figure 1—figure supplement 2D*). All targets that were found to be consistently degraded across multiple large-scale proteomics experiments were validated in those independent validation experiments, providing a high confidence target list (*Figure 1D*).

Eight of the 11 new targets found in the proteomics screen are ZnF proteins (SALL4, ZNF827, ZBTB39, RNF166, ZNF653, ZNF692, ZNF98 and GZF1), and except for RNF166, all contain at least one ZnF domain that has the characteristic features previously described as critical for IMiD-dependent degradation (*An et al., 2017*) (*Figure 1—figure supplement 2E*). We also observe a striking difference in substrate specificity among thalidomide, lenalidomide, and pomalidomide (*Figure 1D*). We find that thalidomide induces robust degradation of the zinc finger transcription factors ZNF692, SALL4, and the ubiquitin ligase RNF166 in cell lines expressing detectable levels of those proteins (*Figure 1D* and *Figure 1—figure supplement 2A*). Lenalidomide results in additional degradation of ZNF827, FAM83F, and RAB28 along with the lenalidomide-specific substrate CSNK1A1. Pomalidomide shows the most pronounced expansion of targets, and in addition induces robust degradation of ZBTB39, ZFP91, DTWD1, and ZNF653. It is interesting to note that DTWD1 is, as CSNK1A1 and GSPT1, another non zinc finger target that was found to be robustly degraded by pomalidomide. Although this expansion of substrates is interesting and may contribute to some of the clinical differences between lenalidomide and pomalidomide, a target causative for teratogenicity would need to be consistently degraded across all IMiDs.

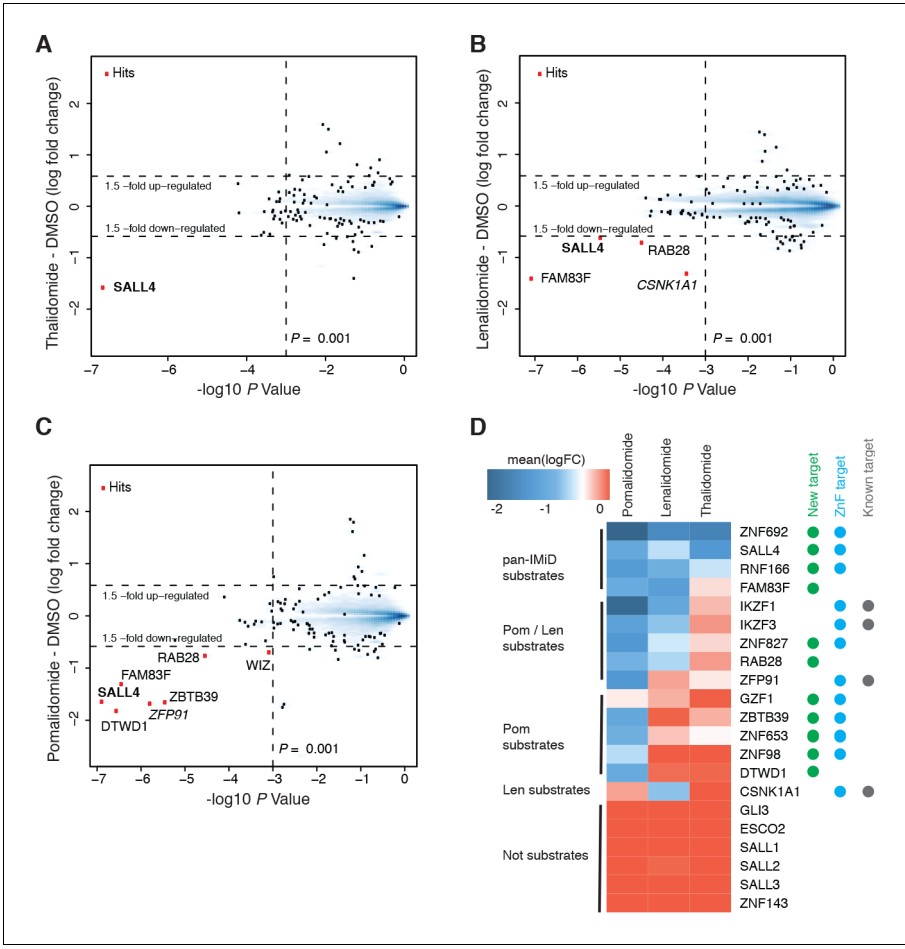

**Figure 1.** Identification of SALL4 as an IMiD-dependent CRL4$^{CRBN}$ substrate. (A–C) Scatter plot depicting identification of IMiD-dependent substrate candidates. H9 human embryonic stem cells (hESC) were treated with 10 μM thalidomide (A), 5 μM lenalidomide (B), 1 μM pomalidomide (C), or DMSO control, and protein abundance was analyzed using TMT quantification mass spectrometry (see Materials and methods for details). Significant changes were assessed by a moderated t-test as implemented in the limma package (*Ritchie et al., 2015*), with the log2 fold change (log2 FC) shown on the y-axis, and negative log$_{10}$ p values on the x-axis (two independent biological replicates for each of the IMiDs, or three independent biological replicates for DMSO). (D) Heatmap displaying the mean log2 FC of the identified IMiD-dependent targets comparing treatment with thalidomide, lenalidomide, and pomalidomide. Mean log2 FC values were derived from averaging across proteomics experiments in four different cell lines (hESC, MM1s, Kelly, SK-N-DZ). The heatmap colors are scaled with blue indicating a decrease in protein abundance (−2 log2 FC) and red indicating no change (0 log2 FC) in protein abundance. Targets newly identified in this study are marked with a green dot, ZnF containing targets with a cyan dot, and previously characterized targets with a gray dot. Substrates are grouped according to their apparent IMiD selectivity in the mass spectrometry-based proteomics. It should be noted that this does not refer to absolute selectivity but rather relative selectivity.

DOI: https://doi.org/10.7554/eLife.38430.003

The following source data and figure supplements are available for figure 1:

**Source data 1.** Output from limma processing of hES cells thalidomide vs. DMSO mass spectrometry experiment.
DOI: https://doi.org/10.7554/eLife.38430.006

**Source data 2.** Output from limma processing of hES cells lenalidomide vs. DMSO mass spectrometry experiment.
DOI: https://doi.org/10.7554/eLife.38430.007

**Source data 3.** Output from limma processing of hES cells pomalidomide vs. DMSO mass spectrometry experiment.
DOI: https://doi.org/10.7554/eLife.38430.008

**Source data 4.** Output from limma processing of Kelly cells thalidomide vs. DMSO mass spectrometry experiment.
DOI: https://doi.org/10.7554/eLife.38430.009

**Source data 5.** Output from limma processing of Kelly cells lenalidomide vs. DMSO mass spectrometry experiment.

*Figure 1 continued on next page*

*Figure 1 continued*

DOI: https://doi.org/10.7554/eLife.38430.010

**Source data 6.** Output from limma processing of Kelly cells pomalidomide vs. DMSO mass spectrometry experiment.
DOI: https://doi.org/10.7554/eLife.38430.011

**Source data 7.** Output from limma processing of MM1s cells thalidomide vs. DMSO mass spectrometry experiment.
DOI: https://doi.org/10.7554/eLife.38430.012

**Source data 8.** Output from limma processing of MM1s cells lenalidomide vs. DMSO mass spectrometry experiment.
DOI: https://doi.org/10.7554/eLife.38430.013

**Source data 9.** Output from limma processing of MM1s cells pomalidomide vs. DMSO mass spectrometry experiment.
DOI: https://doi.org/10.7554/eLife.38430.014

**Source data 10.** Output from limma processing of SK-N-DZ cells pomalidomide vs. DMSO mass spectrometry experiment.
DOI: https://doi.org/10.7554/eLife.38430.015

**Source data 11.** Output from limma processing of SK-N-DZ cells CC-220 vs. DMSO mass spectrometry experiment.
DOI: https://doi.org/10.7554/eLife.38430.016

**Source data 12.** Output from limma processing of SK-N-DZ cells dBET57 vs. DMSO mass spectrometry experiment.
DOI: https://doi.org/10.7554/eLife.38430.017

**Source data 13.** Output from limma processing of SK-N-DZ cells pomalidomide vs. DMSO mass spectrometry experiment.
DOI: https://doi.org/10.7554/eLife.38430.018

**Source data 14.** Output from limma processing of SK-N-DZ cells pomalidomide +MLN4924 vs. DMSO mass spectrometry experiment.
DOI: https://doi.org/10.7554/eLife.38430.019

**Source data 15.** Uncropped immunoblots.
DOI: https://doi.org/10.7554/eLife.38430.020

**Figure supplement 1.** Mass spectrometry profiling of IMiDs.
DOI: https://doi.org/10.7554/eLife.38430.004

**Figure supplement 2.** Extended validation of IMiD-dependent targets.
DOI: https://doi.org/10.7554/eLife.38430.005

## SALL4, a key developmental transcription factor, is a bona fide IMiD-dependent CRL4$^{CRBN}$ target

The robust down-regulation of SALL4, a *spalt*-like developmental transcription factor important for limb development (*Koshiba-Takeuchi et al., 2006*), upon treatment with thalidomide, lenalidomide, and pomalidomide prompted us to further investigate SALL4 as an IMiD-dependent target of CRL4$^{CRBN}$. Strikingly, human genetic research has shown that familial loss of function (LOF) mutations in *SALL4* are causatively linked to the clinical syndromes, Duane Radial Ray syndrome (DRRS) also known as Okihiro syndrome, and mutated in some patients with Holt-Oram syndrome (HOS). Remarkably, both DRRS and HOS have large phenotypic overlaps with thalidomide embryopathy (*Kohlhase et al., 2003*), and this phenotypic resemblance has led to misdiagnosis of patients with *SALL4* mutations as cases of thalidomide embryopathy and the hypothesis that the tbx5/sall4 axis might be involved in thalidomide pathogenesis (*Knobloch and Rüther, 2008*; *Kohlhase et al., 2003*).

Thalidomide embryopathy is characterized not only by phocomelia, but also various other defects (*Table 1*), many of which are specifically recapitulated in syndromes known to originate from heterozygous LOF mutations in *SALL4* (*Kohlhase, 1993*). The penetrance of DRRS in individuals with heterozygous *SALL4* mutations likely exceeds 90% (*Kohlhase, 2004*), and thus partial degradation of SALL4 through IMiD exposure will likely result in similar clinical features observed in DRRS. All currently described *SALL4* mutations are heterozygous LOF mutations, and the absence of homozygous mutations indicates the essentiality of the gene. Accordingly, homozygous deletion of *Sall4* is early embryonic lethal in mice (*Sakaki-Yumoto et al., 2006*). Mice with heterozygous deletion of *Sall4* show a high frequency of miscarriage, while surviving litters show ventricular septal defects and anal stenosis, both phenotypes that are observed in humans with DRRS or thalidomide syndrome (*Sakaki-Yumoto et al., 2006*). Mice carrying a heterozygous *Sall4* genetrap allele show defects in

**Table 1.** Common phenotypes in thalidomide syndrome, Duane Radial Ray syndrome, and Holt-Oram syndrome.

| | Thalidomide syndrome | Duane Radial Ray syndrome | Holt-Oram syndrome |
|---|---|---|---|
| **Upper limbs** | | | |
| | Thumbs | Thumbs | Thumbs |
| | Radius | Radius | Radius |
| | Humerus | Humerus | Humerus |
| | Ulna | Ulna | Ulna |
| | Fingers | Fingers | Fingers |
| **Lower limbs** | | | |
| | Mostly normal lower limbs | Mostly normal lower limbs | |
| | Talipes dislocation | Talipes dislocation | |
| | Hip dislocation | | |
| | Shortening of long bones | | |
| **Ears** | | | |
| | Absence or abnormal pinnae | Abnormal pinnae | |
| | Deafness | Deafness | |
| | Microtia | | |
| **Eyes** | | | |
| | Colobomata | Colobomata | |
| | Microphthalmos | Microphthalmos | |
| | Abduction of the eye | Abduction of the eye | |
| | Duane anomaly | Duane anomaly | |
| **Stature** | | | |
| | Short stature | Postnatal growth retardation | |
| **Heart** | | | |
| | Ventricular septal defects | Ventricular septal defects | Ventricular septal defects |
| | Atrial septal defects | Atrial septal defects | Atrial septal defects |

DOI: https://doi.org/10.7554/eLife.38430.021

heart and limb development, partially reminiscent of patients with DRRS or HOS (*Koshiba-Takeuchi et al., 2006*). Another genetic disorder with a related phenotype is Roberts syndrome, caused by mutations in the ESCO2 gene (*Afifi et al., 2016*). While ESCO2 similarly encodes for a zinc finger protein and is transcriptionally regulated by ZNF143 (*Nishihara et al., 2010*), ESCO2 (as well as ZNF143, SALL1, SALL2, and SALL3) protein levels were found to be unchanged in all of our mass spectrometry experiments despite robust and ubiquitous expression (*Figure 1D*, *Figure 1— figure supplements 1* and *2* and *Figure 1—source data 1–14*).

The remarkable phenotypic overlap of LOF mutations in *SALL4* with thalidomide embryopathy led us to further assess whether thalidomide and related IMiDs directly induce degradation of SALL4 in an IMiD and CRL4$^{CRBN}$-dependent manner. To extend our mass spectrometry findings, we treated H9 hESC with increasing doses of thalidomide, lenalidomide, pomalidomide, or with DMSO as a control, and assessed protein levels of SALL4 by western blot. We observed a dose-dependent decrease in protein levels with all three drugs (*Figure 2A* and *Figure 2—figure supplement 1*), in accordance with IMiD-induced protein degradation. We then used qPCR to confirm that thalidomide treatment does not reduce the level of SALL4 mRNA, but rather upregulates SALL4 mRNA, consistent with the protein-level changes being caused by post-transcriptional effects (*Figure 2—figure supplement 1I*).

We next sought to assess the robustness of SALL4 degradation across different lineages by subjecting a panel of cell lines (Kelly, SK-N-DZ, HEK293T, and H661 cells) to increasing concentrations of thalidomide, lenalidomide, pomalidomide, or DMSO as a control and performed western blot

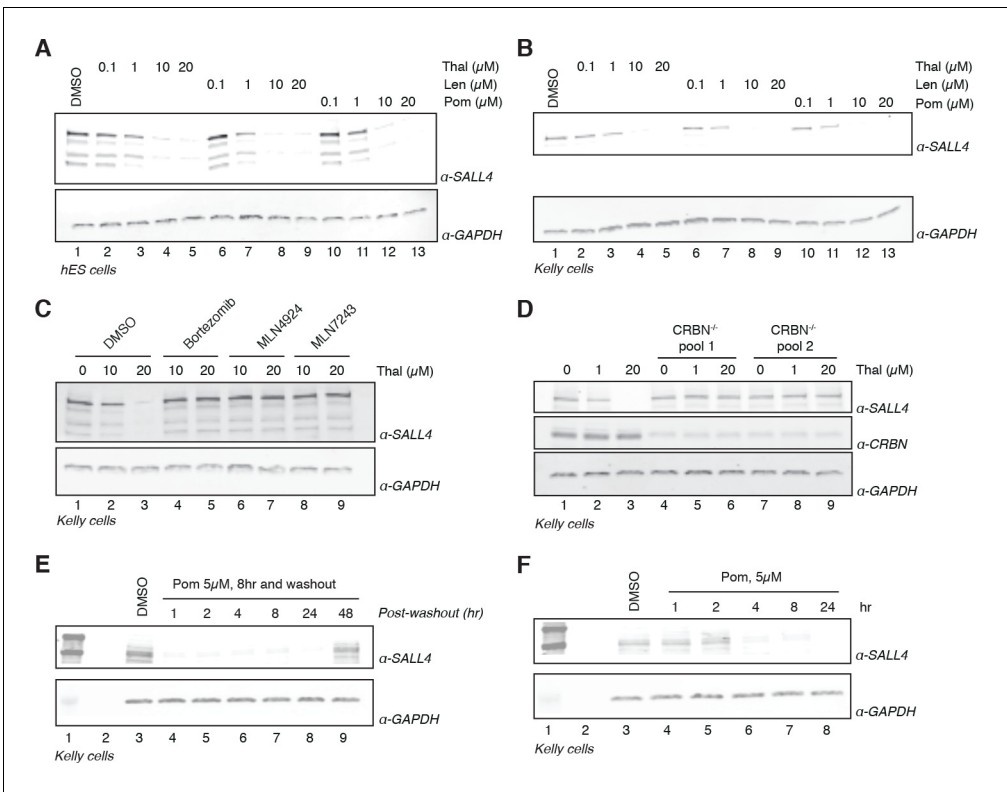

**Figure 2.** Validation of SALL4 as a *bona fide* IMiD-dependent CRL4[CRBN] substrate. (**A**) H9 hESC were treated with increasing concentrations of thalidomide, lenalidomide, pomalidomide, or DMSO as a control. Following 24 h of incubation, SALL4 and GAPDH protein levels were assessed by western blot analysis. (**B**) As in (**A**), but treatment was done in Kelly cells. (**C**) Kelly cells were treated with increasing concentrations of thalidomide and co-treated with 5 µM bortezomib, 5 µM MLN4924, 0.5 µM MLN7243, or DMSO as a control. Following 24 h incubation, SALL4 and GAPDH protein levels were assessed by western blot analysis. (**D**) Parental Kelly cells or two independent pools of CRBN[-/-] Kelly cells were treated with increasing concentrations of thalidomide. Following 24 h incubation, SALL4, CRBN, and GAPDH protein levels were assessed by western blot analysis. (**E**) Kelly cells were treated with 5 µM pomalidomide or DMSO as a control for 8 h, at which point the compound was washed out. Cells were harvested at 1, 2, 4, 8, 24, and 48 h post-washout and SALL4 and GAPDH protein levels were assessed by western blot analysis. (**F**) Kelly cells were treated with 5 µM pomalidomide for 1, 2, 4, 8, and 24 h, or with DMSO as a control. Following time course treatment, SALL4 and GAPDH protein levels were assessed by western blot analysis. One representative experiment is shown in this figure, from three replicates for each of the western blots.
DOI: https://doi.org/10.7554/eLife.38430.022

The following source data and figure supplement are available for figure 2:

**Source data 1.** Uncropped immunoblots.
DOI: https://doi.org/10.7554/eLife.38430.024
**Figure supplement 1.** Extended validation of SALL4.
DOI: https://doi.org/10.7554/eLife.38430.023

---

analysis (*Figure 2B* and *Figure 2—figure supplement 1A–C*). We observed a dose-dependent decrease in SALL4 protein levels with all three IMiD analogs and in all tested cell lines. In accordance with a CRL4[CRBN]-dependent mechanism, the IMiD-induced degradation was abrogated by co-treatment with the proteasome inhibitor bortezomib, the NEDD8 inhibitor MLN4924, or the ubiquitin E1 (UBA1) inhibitor MLN7243 (which blocks all cellular ubiquitination by inhibiting the initial step of the ubiquitin conjugation cascade) (*Figure 2C* and *Figure 2—figure supplement 1D,E*). To further evaluate the CRL4[CRBN]-dependent mechanism, we generated CRBN[-/-] Kelly and HEK293T cells using CRISPR/Cas9 technology and treated the resulting CRBN[-/-] cells and parental cells with increasing concentrations of thalidomide, lenalidomide, or pomalidomide (*Figure 2D* and *Figure 2—figure supplement 1F*). In agreement with the CRBN-dependent mechanism, no degradation of SALL4 was

observed in CRBN$^{-/-}$ cells. Thalidomide has a plasma half-life ($t_{1/2}$) of ~6 to 8 h (~3 h for lenalidomide, ~9 h for pomalidomide) and a maximum plasma concentration ($C_{max}$) of ~5–10 µM (~2.5 µM for lenalidomide, 0.05 µM for pomalidomide) upon a typical dose of 200–400 mg, 25 mg, or 2 mg for thalidomide, lenalidomide, or pomalidomide, respectively (*Chen et al., 2017*; *Hoffmann et al., 2013*; *Teo et al., 2004*). To recapitulate these effects in vitro, we treated Kelly cells with 1 or 5 µM pomalidomide for 8 h, followed by washout of the drug and assessment of time-dependent recovery of SALL4 protein levels (*Figure 2E* and *Figure 2—figure supplement 1G*). Treatment with pomalidomide induces degradation of SALL4 as early as 4 h post treatment (*Figure 2F* and *Figure 2—figure supplement 1H*), which recovered to levels close to pre-treatment level after 48 h post washout (*Figure 2E*), together suggesting that a single dose of IMiD drugs will be sufficient to deplete SALL4 protein levels for >24 h.

## In vitro binding assays confirm IMiD-dependent binding of SALL4 to CRL4$^{CRBN}$

*Bona fide* targets of IMiD-induced degradation typically bind to CRBN (the substrate-recognition domain of the E3 ligase) in vitro in a compound-dependent manner. Thus, we sought to test whether SALL4 binds to CRBN and to map the ZnF domain required for binding using purified recombinant proteins. Based on conserved features among IMiD-sensitive ZnF domains (*Figure 3A*, C–x(2)-C-G motif within the canonical $C_2H_2$ zinc finger motif), the second (SALL4$_{ZnF2}$) and fourth (SALL4$_{ZnF4}$) ZnF domains of SALL4 (aa 410–433, and aa 594–616, respectively) were identified as candidate degrons for IMiD-induced binding. We expressed, purified, biotinylated, and subjected these ZnF domains to in vitro CRBN binding assays (*An et al., 2017*; *Petzold et al., 2016*). We observed dose-dependent binding between SALL4$_{ZnF2}$ or SALL4$_{ZnF4}$ and CRBN similar to that described for IKZF1/3 and ZFP91, albeit with reduced apparent affinity for SALL4$_{ZnF4}$ (*Figure 3B,C*) (*Petzold et al., 2016*). To estimate apparent affinities ($K_{D(app)}$) we titrated bodipy-FL labelled DDB1ΔB-CRBN to biotinylated SALL4$_{ZnF2}$, or SALL4$_{ZnF4}$ at 100 nM with saturating concentrations of IMiDs (50 µM) and measured the affinity by TR-FRET (*Figure 3D* and *Figure 3—figure supplement 1A,B*), which confirmed the weak affinity of SALL4$_{ZnF4}$. However, we noticed that a construct spanning ZnF1 and ZnF2 of SALL4 (SALL4$_{ZnF1-2}$) exhibited even tighter binding to CRBN (*Figure 3D* and *Figure 3—figure supplement 1A,B*) and enhanced dose-dependent complex formation in TR-FRET (*Figure 3E*). These findings suggest that multiple zinc finger domains of SALL4 contribute to binding, and may result in multivalent recruitment to CRBN in vivo. However, the strength of the interaction with ZnF4 is unlikely to be sufficient for degradation in cells, and moreover, the rank order of Pom >Thal >> Len in binding observed with ZnF2 is in accordance with the cellular potency in degradation of SALL4, suggesting that ZnF2 is the critical ZnF domain for SALL4 degradation. We confirmed the specificity of the SALL4$_{ZnF2}$ interaction by introducing a point mutation to glycine 416 (G416), the residue critical for IMiD-dependent binding to CRBN (*Petzold et al., 2016*). Mutations to alanine (G416A) rendered SALL4$_{ZnF2}$ resistant to IMiD-dependent binding to CRBN (*Figure 3F* and *Figure 3—figure supplement 1C,D*). Mutating glutamine 595 (Q595) in SALL4$_{ZnF4}$, another residue previously shown to be critical for IMiD-dependent CRBN binding in the ZnF domains of IKZF1/3, impaired IMiD-dependent binding (*Figure 3—figure supplement 1E*), confirming the specificity of the interaction despite the weak binding affinity. As we observed increased affinity of the tandem-ZnF construct SALL4$_{ZnF1-2}$ compared with the single SALL4$_{ZnF2}$, we sought to test whether ZnF1 was sufficient for binding. We introduced the G416N mutation in ZnF2 or a S388N mutation in ZnF1 into the SALL4$_{ZnF1-2}$ construct (S388 is the ZnF1 sequence equivalent of ZnF2 G416; ZnF1-2: C-x-x-C-S/G) and performed CRBN binding assays. G416N, but not S388N, fully abrogated IMiD-dependent binding of SALL4$_{ZnF1-2}$ to CRBN (*Figure 3—figure supplement 1F–I*), confirming the strict dependence on the ZnF2 interaction with CRBN. To test whether the second zinc finger of SALL4 is critical for IMiD-induced degradation in cells, we introduced G416A and G416N mutations into Flag-tagged full-length SALL4. When expressed in Kelly cells, the parental wild-type Flag-SALL4 was readily degraded by thalidomide treatment (*Figure 3G*). Similarly, Flag-tagged SALL4 with G600A or G600N mutations in ZnF4 were also shown to be readily degraded with thalidomide treatment, suggesting that SALL4$_{ZnF4}$ is dispensable for binding and subsequent degradation (*Figure 3G*). Finally, the two conservative mutations in ZnF2 (G416A or G416N), both known to specifically disrupt binding to CRBN while maintaining the overall zinc finger fold (*Petzold et al., 2016*), rendered SALL4 stable under these treatment conditions, demonstrating that SALL4$_{ZnF2}$ is necessary for CRL4$^{CRBN}$-mediated

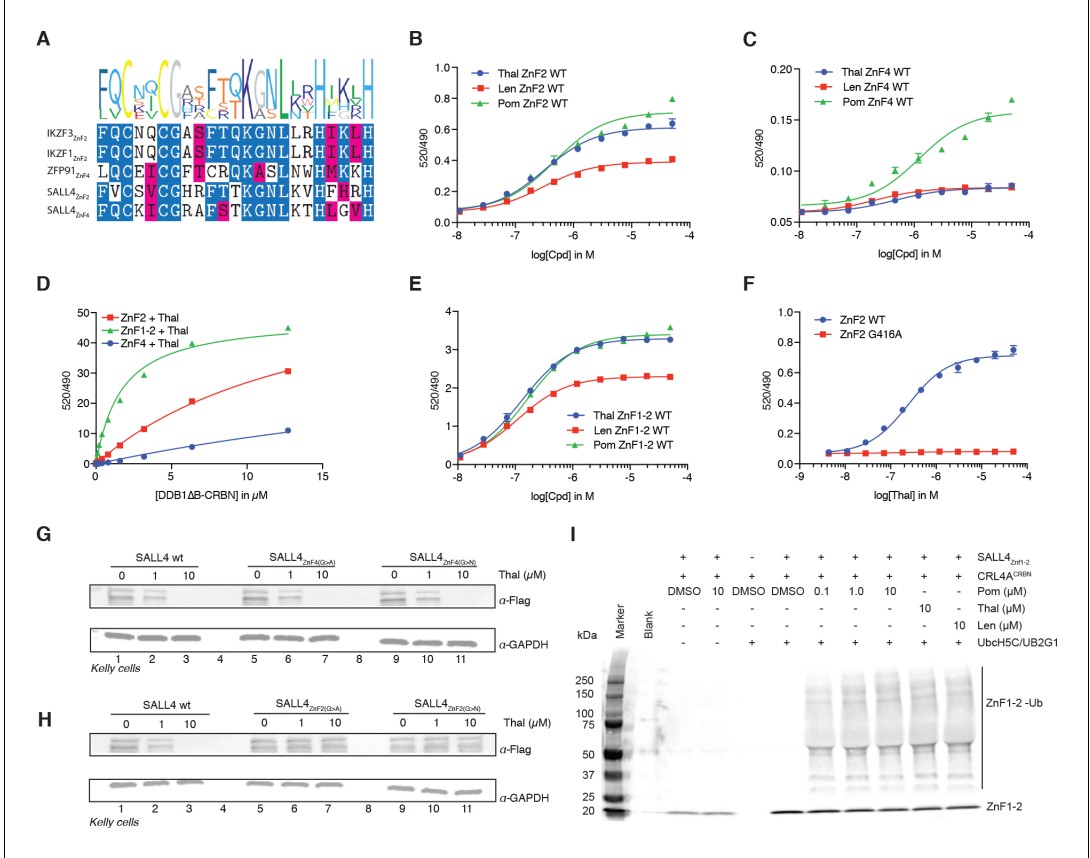

**Figure 3.** SALL4 ZnF2 is the zinc finger responsible for IMiD-dependent binding to CRL4$^{CRBN}$. (**A**) Multiple sequence alignment of the validated 'degrons' from known IMiD-dependent zinc finger substrates, along with the two candidate zinc finger degrons from SALL4. (**B**) TR-FRET: titration of IMiD (thalidomide, lenalidomide and pomalidomide) to DDB1ΔB-CRBN$_{Spy-BodipyFL}$ at 200 nM, hsSALL4$_{ZnF2}$ at 100 nM, and terbium-streptavidin at 4 nM. (**C**) As in (**B**), but with hsSALL4$_{ZnF4}$ with DDB1ΔB-CRBN$_{Spy-BodipyFL}$ at 1 μM. (**D**) TR-FRET: titration of DDB1ΔB-CRBN$_{Spy-BodipyFL}$ to biotinylated hsSALL4$_{ZnF2}$, hsSALL4$_{ZnF1-2}$, or hsSALL4$_{ZnF4}$ at 100 nM and terbium-streptavidin at 4 nM in the presence of 50 μM thalidomide. (**E**) As in (**B**), but with hsSALL4$_{ZnF1-2}$. (**F**) As in (**B**), but with hsSALL4$_{ZnF2}$ and hsSALL4$_{ZnF2}^{G416A}$ mutant as thalidomide titration. (**G**) Kelly cells transiently transfected with Flag-hsSALL4$^{WT}$, Flag-hsSALL4$^{G600A}$, or hsSALL4$^{G600N}$ were treated with increasing concentrations of thalidomide or DMSO as a control. Following 24 h of incubation, SALL4 (α-Flag) and GAPDH protein levels were assessed by western blot analysis (one representative experiment is shown out of three replicates. (**H**) As in (**G**), but with Flag-hsSALL4$^{WT}$, Flag-hsSALL4$^{G416A}$, or Flag-hsSALL4$^{G416N}$. (**I**) In vitro ubiquitination of biotinylated hsSALL4$_{ZnF1-2}$ by CRL4$^{CRBN}$ in the presence of thalidomide (10 μM), lenalidomide (10 μM), or pomalidomide (0.1, 1 and 10 μM), or DMSO as a control.

DOI: https://doi.org/10.7554/eLife.38430.025

The following source data and figure supplement are available for figure 3:

**Source data 1.** Uncropped immunoblots.

DOI: https://doi.org/10.7554/eLife.38430.027

**Figure supplement 1.** Biochemical characterization of SALL4 binding to CRBN.

DOI: https://doi.org/10.7554/eLife.38430.026

degradation of SALL4 in cells (*Figure 3H*). In vitro ubiquitination assays further confirm that SALL4$_{ZnF1-2}$ is ubiquitinated by CRL4$^{CRBN}$ in an IMiD-dependent fashion (*Figure 3I*). Together, our cellular and biochemical data establish SALL4 as a *bona fide* IMiD-dependent target of CRL4$^{CRBN}$, and demonstrate that the second zinc finger is necessary for IMiD-dependent degradation, while the tandem array of ZnF1-2 further strengthens the interaction in vitro.

## Species-specific teratogenicity is a result of genetic differences in both CRBN and SALL4

One characteristic feature of IMiD phenotypes is the absence of defining limb deformities following administration to pregnant rodents, which contributed to the initial approval by regulatory agencies in Europe. In contrast, many non-human primates exhibit phenotypes that mimic the human

syndrome (*Neubert et al., 1988*; *Smith et al., 1965*; *Vickers, 1967*). These remarkable species-specific phenotypes have historically complicated studies of thalidomide embryopathies, and suggest a genetic difference between these species that would abrogate the detrimental effects of thalidomide. Mouse Crbn harbors a critical polymorphism (*Figure 4A,B* and *Figure 5*) that prevents IMiD-dependent degradation of ZnF substrates and CSNK1A1 (*Krönke et al., 2015*), which could explain the absence of a SALL4-dependent phenotype in mice. Mice and rats (both insensitive to thalidomide embryopathies) harbor an isoleucine at CRBN position 388 (residue 388 refers to the human CRBN sequence). In contrast, sensitive primates have a valine in position 388 that is necessary for CRL4$^{CRBN}$ to bind, ubiquitinate, and subsequently degrade ZnF substrates (*Figure 4A,B* and *Figure 5A*). Consistent with this concept, treatment of mouse embryonic stem cells (mESC) with increasing concentrations of thalidomide or pomalidomide does not promote degradation of mmSALL4 (*Figure 4C* and *Figure 4—figure supplement 1A*), and introducing a V388I mutation into hsCRBN renders the protein less effective to bind to SALL4 in vitro (*Figure 4B*). We thus asked whether ectopic expression of hsCRBN in mouse cells would lead to IMiD-induced degradation of mmSALL4, similar to what had been observed for CSNK1A1, and could hence render mice sensitive to IMiD-induced birth defects. Expression of hsCRBN in mouse cells, while sensitizing cells to degradation of IMiD targets such as mmIKZF1/3, mmCSNK1A1 (*Krönke et al., 2015*), mmZFP91, or mmGZF1 (*Figure 4D,E*), does not result in degradation of mmSALL4 (*Figure 4F*). To test whether a fully human CRBN in a human cell background would be sufficient to induce SALL4 degradation, we introduced hsSALL4, or mmSALL4 into human cells (Kelly cells) and found that while ectopically expressed hsSALL4 is readily degraded upon IMiD treatment, mmSALL4 is unaffected even at arbitrarily high doses of IMiDs (*Figure 4G* and *Figure 4—figure supplement 1B,C*). Sequence analysis reveals that mice and zebrafish have critical mutations in the ZnF2 domain of SALL4 (*Figure 5B*), which abrogate binding to hsCRBN in vitro (*Figure 4H*), and render mmSALL4 and drSALL4 insensitive to IMiD-mediated degradation in cells (*Figure 4G,I* and *Figure 4—figure supplement 1C*). In line with these findings, mice harboring a homozygous CRBN I391V knock-in allele, despite exhibiting degradation of mmIKZF1/3, mmZFP91, and mmCSNK1A1 *Fink et al., 2018*, show increased miscarriage upon IMiD treatment compared with control mice; however, they do not exhibit IMiD-induced embryopathies resembling the human phenotype *Fink et al., 2018* . We next sought to test whether exchange of the mmSALL4 ZnF2 domain for the hsSALL4 ZnF2 domain would be sufficient to enable mmSALL4 degradation in a human cell line (Kelly cells). Strikingly, through the five amino acid substitutions required to 'humanize' the mmSALL4 ZnF2 domain, we were able to induce thalidomide-mediated mouse SALL4 degradation in a human cell line (*Figure 4G*).

The observation that SALL4 degradation depends on both the sequence of SALL4 (zinc finger 2 differs between human and rodents), and the sequence of CRBN, supports a genetic cause for the species-specific effects, and highlights the complexities of modelling teratogenic adverse effects of IMiDs in murine and other animal models (*Sakaki-Yumoto et al., 2006*) (*Figure 5A–C*). Of note, the only non-human primate known to be insensitive to thalidomide-induced embryopathies, the greater bush baby, also harbors an isoleucine in the critical CRBN V388 position (*Butler, 1977*), while all sensitive non-human primates and rabbits harbor the conserved valine (*Figure 5A*). We thus show that species can be rendered resistant by mutations in CRBN, SALL4, or both, and hence our data suggest that thalidomide embryopathy is primarily a human disease (with some non-human primates, and rabbits more closely resembling the phenotypes), and thus explain the historic observation that modelling thalidomide embryopathies in animals is challenging. We note that zebrafish and chicken both contain an Ile in the V388 position; however, these were reported to exhibit defects to limb/fin formation upon exposure to thalidomide or knock-down of *Crbn* (*Eichner et al., 2016*; *Ito et al., 2010*), partially resembling thalidomide-induced defects. These findings are in contrast with the observations in higher eukaryotes, as *Crbn* knock-out mice have been reported to exhibit normal morphology (*Lee et al., 2013*), and children harboring a homozygous C391R mutation in CRBN (C391 is a structural cysteine coordinating the zinc in the thalidomide-binding domain of CRBN and we failed to produce any protein from a C391R cDNA), a loss of function mutation, were born without characteristic birth defects but exhibited severe neurological defects (*Sheereen et al., 2017*). Whether the phenotypes in zebrafish and chicken are a result of species-specific downstream pathways or the high dose (400 µM) and direct application of thalidomide to the limb buds (*Ito et al., 2010*), which both could result in off-target effects, remains to be shown. The plasma concentration of thalidomide in humans is, however, unlikely to exceed 10 µM (*Bai et al., 2013*; *Dahut et al.,*

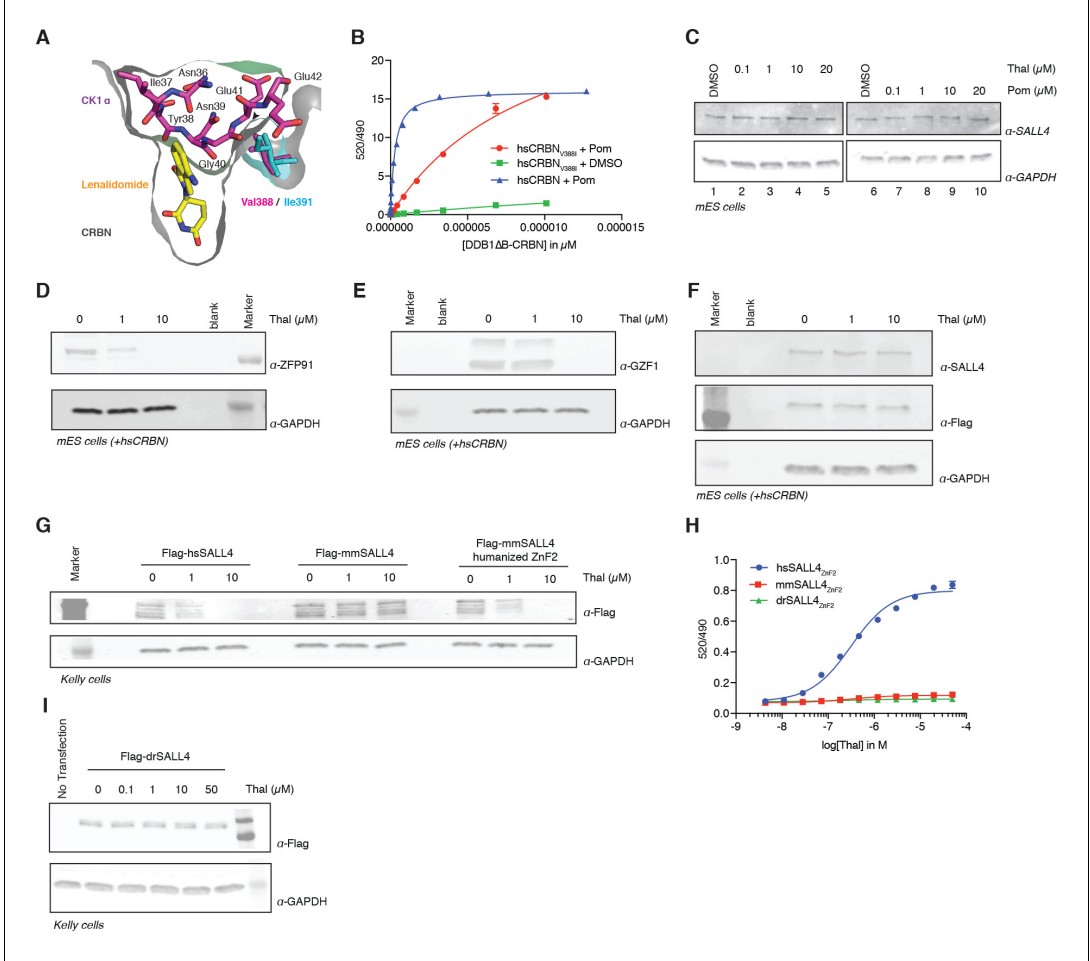

**Figure 4.** Identification of the sequence differences in the IMiD-dependent binding region of both CRBN and SALL4 in specific species. (A) Close-up view of the beta-hairpin loop region of Ck1a (CSNK1A1) interacting with CRBN and lenalidomide (PDB: 5fqd) highlighting the additional bulkiness of the V388I mutation (PDB: 4ci1) present in mouse and rat CRBN. CSNK1A1 and lenalidomide are depicted as stick representations in magenta and yellow, respectively, the Ile391 of mouse CRBN corresponding to human Val388 is depicted as a stick representation in cyan, and CRBN is depicted as a surface representation. (B) TR-FRET: titration of DDB1ΔB-hsCRBN$_{Spy-BodipyFL}$, or DDB1ΔB-hsCRBN$^{V388I}_{Spy-BodipyFL}$ to biotinylated hsSALL4$_{ZnF1-2}$ at 100 nM, and terbium-streptavidin at 4 nM in the presence of 50 μM pomalidomide or DMSO. (C) mES cells were treated with increasing concentrations of thalidomide and pomalidomide or DMSO as a control. Following 24 h of incubation, SALL4 and GAPDH protein levels were assessed by western blot analysis. (D) mES cells constitutively expressing Flag-hsCRBN were treated with increasing concentrations of thalidomide. Following 24 h of incubation, ZFP91 and GAPDH protein levels were assessed by western blot analysis. (E) As in (C), but measuring GZF1 and GAPDH protein levels. (F) As in (C), but measuring SALL4, hsCRBN (α-Flag), and GAPDH protein levels. (G) Kelly cells were transiently transfected with Flag-hsSALL4, Flag-mmSALL4, or Flag-mmSALL4 containing a humanized ZnF2 (Y415F, P418S, I419V, L430F, Q435H), and treated with increasing concentrations of thalidomide. Following 24 h of incubation, hsSALL4, mmSALL4, humanized mmSALL4 (α-Flag), and GAPDH protein levels were assessed by western blot analysis. (H) TR-FRET: titration of thalidomide to DDB1ΔB-CRBN$_{Spy-BodipyFL}$ at 200 nM, hsSALL4$_{ZnF2}$, mmSALL4$_{ZnF2}$, or drSALL4$_{ZnF2}$ all at 100 nM, and terbium-streptavidin at 4 nM. Data are presented as means ± s.d. (n = 3). (I) As in (G), but with Flag-drSALL4.
DOI: https://doi.org/10.7554/eLife.38430.028

The following source data and figure supplement are available for figure 4:

**Source data 1.** Uncropped immunoblots.
DOI: https://doi.org/10.7554/eLife.38430.030
**Figure supplement 1.** Species-specific effects.
DOI: https://doi.org/10.7554/eLife.38430.029

*2009*), a concentration that results in effective degradation of SALL4, but is 40 times below the dose found to be teratogenic in chicken and zebrafish embryos. While we do not observe degradation of mmSALL4 or drSALL4 upon high-dose exposure, we cannot rule out that such high doses will induce degradation of other ZnF targets in zebrafish or chicken, which could potentially result in the

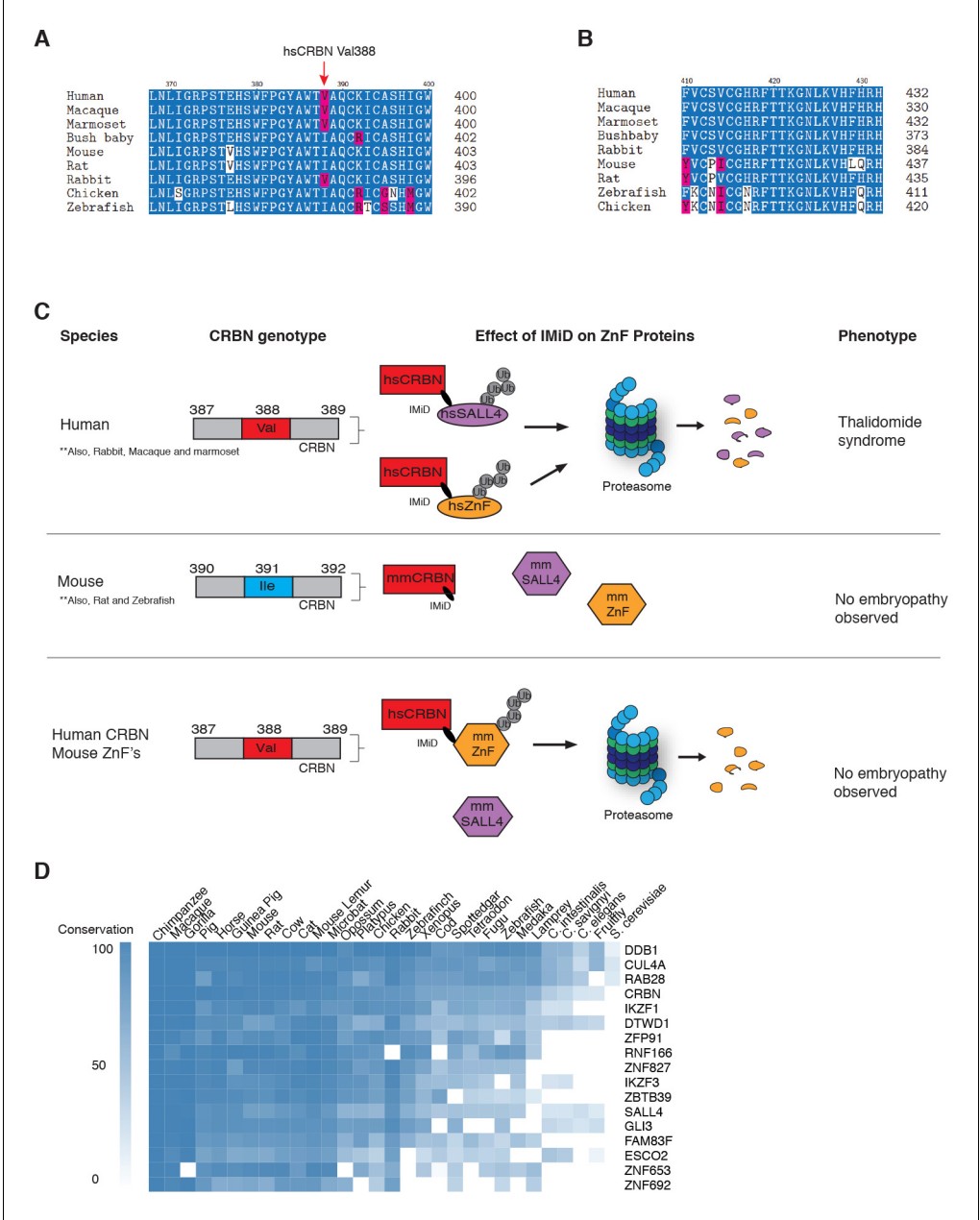

**Figure 5.** Sequence differences in the IMiD-dependent binding region of both CRBN and SALL4 interfere with ternary complex formation in specific species. (**A**) A multiple sequence alignment of the region of CRBN critical for IMiD mediated ZnF binding from human, bush baby mouse, rat, macaque, marmoset, and rabbit is shown, highlighting the V388I polymorphism. (**B**) A multiple sequence alignment of SALL4$_{ZnF2}$ from human, macaque, marmoset, bush baby, rabbit, mouse, rat, zebrafish, and chicken, highlighting the differences in sequence across species. (**C**) Schematic summary of species-specific effects of IMiD treatment on ZnF degradation and relationship to thalidomide syndrome phenotype. The top panel depicts sensitive species: hsCRBN$^{V388}$ is capable of IMiD-dependent binding, ubiquitination, and subsequent degradation of hsSALL4 and hsZnF targets, and thalidomide embryopathy is observed. The middle panel depicts insensitive species: mmCRBN$^{I391}$ is capable of binding IMiDs, but not binding mmSALL4 and mmZnF targets, and no embryopathy is observed. The bottom panel depicts humanizing CRBN as ineffective for inducing the phenotype: hsCRBN$^{V388}$ is capable of IMiD-dependent ubiquitination and subsequent degradation of mmZnF proteins, but not mmSALL4, and the embryopathy is not observed. These data are consistent with a 'double protection' mechanism caused by mutations in both CRBN and SALL4 preventing IMiD-dependent binding and subsequent degradation in insensitive species. (**D**) Heatmap

*Figure 5 continued on next page*

*Figure 5 continued*
comparing the sequence conservation of IMiD-dependent targets across 30 different species. High conservation is displayed as blue and low conservation is displayed as white.
DOI: https://doi.org/10.7554/eLife.38430.031

observed phenotypes. In fact, we show that IMiDs lead to degradation of multiple ZnF transcription factors, a class of proteins known to evolve very rapidly (*Schmitges et al., 2016*), and it is likely that IMiDs will exhibit species-specific effects. Sequence analysis shows that IMiD-dependent ZnF targets such as SALL4, ZNF653, ZNF692, or ZBTB39, as well as other known genetic causes of limb defects in ZnF transcription factors, such as ESCO2, are highly divergent even in higher eukaryotes (*Figure 5D*).

## Discussion

We show that thalidomide, lenalidomide, and pomalidomide all induce degradation of SALL4, which has been causatively linked to the most characteristic and common birth defects of the limbs and inner organs by human genetics. While other targets of thalidomide, such as CSNK1A1 for lenalidomide or GZF1, ZBTB39 for pomalidomide, may contribute to the pleiotropic developmental conditions observed upon thalidomide exposure, SALL4 is consistently degraded across all IMiDs and human geneticists associate heterozygous loss of SALL4 with human developmental syndromes that largely phenocopy thalidomide syndrome. Moreover, from the targets degraded across IMiDs, IKZF1/3 have been shown to be non-causative for birth defects, RNF166 is a ubiquitin ligase involved in autophagy (*Heath et al., 2016*), and ZNF692 knock-out mice do not exhibit a teratogenic phenotype (*International Mouse Phenotyping Consortium et al., 2016*). While only genetic studies in non-human primates or rabbits can provide the ultimate molecular role of SALL4 and other targets in thalidomide embryopathies, the known functions of SALL4 are consistent with a potential role in thalidomide embryopathies.

The polypharmacology of IMiDs (most notably pomalidomide), together with the size and rapid evolution of the $C_2H_2$ family of zinc finger transcription factors (*Figure 5D*), which results in most $C_2H_2$ zinc finger transcription factors being highly species-specific (*Najafabadi et al., 2015*; *Schmitges et al., 2016*), help to explain the pleiotropic effects of IMiDs, which still remain largely understudied. Thalidomide embryopathies thus represent a case in which animal studies fall short, and it is likely that the clinical features of IMiD efficacy as well as adverse effects, are a result of induced degradation of multiple $C_2H_2$ zinc finger transcription factors. For example, we see some degree of degradation for GZF1, another $C_2H_2$ transcription factor, while GZF1 is unlikely to cause the defining birth defects of thalidomide, mutations in GZF1 have been associated with joint laxity and short stature, which are both also found in thalidomide-affected children (*Patel et al., 2017*). We also note that CRBN expression levels influence the efficacy of IMiDs in inducing protein degradation, and it is conceivable that these contribute to a certain degree of tissue selectivity of IMiD effects, which for example, could increase the therapeutic index in MM as hematopoietic lineages tend to have high levels of CRBN.

Thalidomide teratogenicity was a severe and widespread public health tragedy, affecting more than 10,000 individuals, and the aftermath has shaped many of the current drug regulatory procedures. Our findings that thalidomide and its derivatives induce degradation of SALL4, provide a direct link to genetic disorders of SALL4 deficiency, which phenocopy many of the teratogenic effects of thalidomide. While other effects of thalidomide, such as anti-angiogenic properties may contribute to birth defects, degradation of SALL4 is likely to contribute to birth defects. These findings can inform the development of new compounds that induce CRBN-dependent degradation of disease-relevant proteins but avoid degradation of developmental transcription factors such as SALL4, and thus have the potential for therapeutic efficacy without the risk of teratogenicity, a defining feature of this class of drugs. This is further relevant to the development of thalidomide-derived bifunctional small molecule degraders (commonly referred to as PROTACs) (*Raina and Crews, 2017*), as we show that IMiD-based PROTACs (and novel IMiD derivatives such as CC-220) can be effective inducers of ZnF targets including SALL4 degradation (*Figure 1—figure supplement 1C*). Lastly, the surprising expansion in substrate repertoire for pomalidomide, suggests that IMiDs

exhibit a large degree of polypharmacology contributing to both efficacy and adverse effects. Transcription factors, and specifically $C_2H_2$ zinc fingers are highly divergent between species, and hence IMiDs and related compounds are likely to exhibit species-specific effects by virtue of their mode of action. In turn, the discovery that IMiDs target an unanticipated large set of $C_2H_2$ zinc finger proteins with significant differences among thalidomide, lenalidomide, pomalidomide, and CC-220, suggests that this chemical scaffold holds the potential to target one of the largest families of human transcription factors.

# Materials and methods

## Key resources table

| Reagent type (species) or resource | Designation | Source or reference | Identifiers | Additional information |
|---|---|---|---|---|
| Gene (*H. sapiens*) | CRBN | *Fischer et al. (2014)* | Gene ID: 51185 | |
| Gene (*M. musculus*) | CRBN | Dr. Ben Ebert (Brigham and Womens Hospital, Dana Farber Cancer Institute) | Gene ID: 58799 | |
| Gene (*H. sapiens*) | SALL4 | IDT | Gene ID: 57167 | |
| Gene (*H. sapiens*) | DDB1ΔB | *Petzold et al. (2016)* | Gene ID: 1642 | |
| Gene (*M. musculus*) | SALL4 | IDT | Gene ID: 99377 | |
| Gene (*D. rerio*) | SALL4 | IDT | Gene ID: 572527 | |
| Cell line (*H. sapiens*) | H9 hES cells | Dr. Wade Harper (Harvard Medical School) | RRID:CVCL_9773 | |
| Cell line (*H. sapiens*) | Kelly Cells | Dr. Nathanael Gray (Dana Farber Cancer Institute, Harvard Medical School) | RRID:CVCL_2092 | |
| Cell line (*H. sapiens*) | SK-N-DZ cells | ATCC | RRID:CVCL_1701; CRL-2149 | |
| Cell line (*H. sapiens*) | MM1s cells | ATCC | RRID:CVCL_8792; CRL-2974 | |
| Cell line (*H. sapiens*) | H661 cells | ATCC | RRID:CVCL_1577; HTB-183 | |
| Cell line (*H. sapiens*) | HEK293T cells | ATCC | RRID:CVCL_0063; CRL-3216 | |
| Cell line (*M. musculus*) | TC1 mESC cells | Dr. Richard Gregory (Boston Childrens Hospital, Harvard Medical School) | RRID:CVCL_M350 | |
| Cell line (*T. ni*) | High Five insect cells | Thermo Fisher Scientific | RRID:CVCL_C190; B85502 | |
| Chemical compound, drug | Thalidomide | MedChemExpress | HY-14658 | |
| Chemical compound, drug | Lenalidomide | MedChemExpress | HY-A0003 | |
| Chemical compound, drug | Pomalidomide | MedChemExpress | HY-10984 | |
| Chemical compound, drug | CC-220 | MedChemExpress | HY-101291 | |
| Chemical compound, drug | CC-885 | Cayman chemical | 19966 | |
| Chemical compound, drug | dBET57 | *Nowak et al. (2018)* | | |
| Chemical compound, drug | Bortezomib | MedChemExpress | HY-10227 | |
| Chemical compound, drug | MLN4924 | MedChemExpress | HY-70062 | |
| Chemical compound, drug | MLN7243 | Active Biochem | A1384 | |

*Continued on next page*

Continued

| Reagent type (species) or resource | Designation | Source or reference | Identifiers | Additional information |
|---|---|---|---|---|
| Recombinant DNA reagent | pCDH-MSCV (PGK promoter plasmid) | Dr. Ben Ebert (Brigham and Womens Hospital, Dana Farber Cancer Institute) | | |
| Recombinant DNA reagent | pNTM (CMV promoter plasmid) | Dr. Nicolas Thomä, FMI, Switzerland | | |
| Recombinant DNA reagent | pAC8 (Polyhedrin promoter plasmid) | Dr. Nicolas Thomä, FMI, Switzerland | | |
| Peptide, recombinant protein | hsHis6-3C-Spy-CRBN | *Nowak et al. (2018)* | | |
| Peptide, recombinant protein | hsHis6-3C-Spy-CRBN_V388I | This study | | |
| Peptide, recombinant protein | hsStrep-BirA-SALL4 (590–618) | This study | | |
| Peptide, recombinant protein | hsStrep-BirA-SALL4_Q595H (590-618) | This study | | |
| Peptide, recombinant protein | hsStrep-BirA-SALL4 (378–438) | This study | | |
| Peptide, recombinant protein | hsStrep-BirA-SALL4 (402–436) | This study | | |
| Peptide, recombinant protein | mmStrep-BirA-SALL4 (593–627) | This study | | |
| Peptide, recombinant protein | drStrep-BirA-SALL4 (583–617) | This study | | |
| Peptide, recombinant protein | SpyCatcher S50C | *Nowak et al. (2018)* | | |
| Peptide, recombinant protein | His-hsDDB1(1–1140)-His-hsCUL4A (38-759)-His-mmRBX1(12–108) (CRL4-CRBN) | *Fischer et al. (2011)* | | |
| Peptide, recombinant protein | Ubiquitin | Boston Biochem | U-100H | |
| Peptide, recombinant protein | His-E1 | Boston Biochem | E-304 | |
| Peptide, recombinant protein | UBE2G1 | Boston Biochem | E2-700 | |
| Peptide, recombinant protein | UbcH5c | Boston Biochem | E2-627 | |
| Antibody | Mouse anti-SALL4 | abcam | RRID:AB_2183366; ab57577 | WB (1:250) |
| Antibody | Rabbit anti-SALL4 - chip grade | abcam | RRID:AB_777810; ab29112 | WB (1:250) |
| Antibody | Rabbit anti-DTWD1 | Sigma Aldrich | RRID:AB_2677903; HPA042214 | WB (1:500) |
| Antibody | Mouse anti-FLAG M2 | Sigma Aldrich | RRID:AB_262044; F1804 | WB (1:1000) |
| Antibody | Rabbit anti-CRBN | Novus Biologicals | RRID:AB_11037820; NBP1-91810 | WB (1:500) |
| Antibody | Rabbit anti-GZF1 | Thermo Fisher Scientific | RRID:AB_2551727; PA534375 | WB (1:500) |
| Antibody | Mouse anti-GAPDH | Sigma Aldrich | RRID:AB_1078991; G8795 | WB (1:10,000) |
| Antibody | IRDye680 Donkey anti-mouse IgG | LiCor | RRID:AB_10953628; 92668072 | WB (1:10,000) |
| Antibody | IRDye800 Goat anti-rabbit | LiCor | RRID:AB_621843; 92632211 | WB (1:10,000) |
| Antibody | Rabbit anti-Strep-Tag II | abcam | RRID:AB_1524455; ab76949 | WB (1:10,000) |
| Antibody | anti-Strep-Tag II HRP conjugate | Millipore | RRID:AB_10806716; 71591 | WB (1:10,000) |
| Antibody | anti-Mouse IgG HRP conjugate | Cell Signalling | RRID:AB_330924; 7076 | WB (1:10,000) |
| Other | Amersham ECL prime western blot reagent | GE healthcare | RPN2232 | |

*Continued on next page*

*Continued*

| Reagent type (species) or resource | Designation | Source or reference | Identifiers | Additional information |
|---|---|---|---|---|
| Other | BODIPY-FL-Maleimide | Thermo Fisher Scientific | B10250 | |
| Other | Tb streptavidin | Invitrogen | LSPV3966 | |
| Other | TMT 10-plex labels | Thermo Fisher Scientific | 90406 | |
| Other | Lipofectamine 2000 | Invitrogen | 11668019 | |

## Compounds, enzymes, and antibodies

Thalidomide (HY-14658, MedChemExpress), lenalidomide (HY-A0003, MedChemExpress), pomalidomide (HY-10984, MedChemExpress), CC-220 (HY-101291, MedChemExpress), CC-885 (19966, Cayman chemical), dBET57 (*Nowak et al., 2018*), bortezomib (HY-10227, MedChemExpress), MLN4924 (HY-70062, MedChemExpress), and MLN7243 (A1384, Active Biochem) were purchased from the indicated vendors and subjected to in-house LC-MS for quality control.

HEK293T, SK-N-DZ, MM1s, and H661 were purchased from ATCC and cultured according to ATCC instructions. H9 hESC, mESC, and Kelly cells were kindly provided by the labs of J. Wade Harper (HMS), Richard I. Gregory (TCH/HMS), and Nathanael Gray (DFCI/HMS), respectively. Sequencing grade modified trypsin (V5111) was purchased from Promega (Promega, USA) and mass spectrometry grade lysyl endopeptidase from Wako (Wako Pure Chemicals, Japan). Primary and secondary antibodies used included anti-SALL4 at 1:250 dilution (ab57577, abcam – found reactive for human SALL4), anti-SALL4 chip grade at 1:250 dilution (ab29112, abcam – found reactive for mouse Sall4), anti-DTWD1 1:500 (HPA042214, Sigma), anti-Flag 1:1000 (F1804, Sigma), anti-CRBN 1:500 (NBP1-91810, Novus Biologicals), anti-GZF1 at 1:500 (PA534375, Thermo Fisher Scientific), anti-GAPDH at 1:10,000 dilution (G8795, Sigma), IRDye680 Donkey anti-mouse at 1:10,000 dilution (926–68072, LiCor), IRDye800 Goat anti-rabbit at 1:10,000 dilution (926–32211, LiCor) and rabbit anti-Strep-Tag II antibody at 1:10,000 (ab76949, Abcam), anti-mouse IgG HRP-linked Antibody at 1:10,000 dilution (7076, Cell Signaling), Amersham ECL Prime Western Blotting Detection Reagent (RPN2232, GE).

## Cell culture

HEK293T cells were cultured in DMEM supplemented with 10% dialyzed fetal bovine serum (FBS) and 2 mM L-glutamine. SK-N-DZ cells were cultured in DMEM supplemented with 10% dialyzed FBS, 0.1 mM non-essential amino acids (NEAA), and 2 mM L-glutamine. H661, MM1s, and Kelly cells were cultured in RPMI1640 supplemented with 10% dialyzed FBS. H9 hESC cells were cultured in Essential 8 (Gibco) media on Matrigel-coated nunc tissue culture plates. TC1 mouse embryonic stem cells (mESCs) were adapted to gelatin cultures and fed with KO-DMEM (Gibco) supplemented with 15% stem cell-qualified fetal bovine serum (FBS, Gemini), 2 mM L-glutamine (Gibco), 20 mM HEPES (Gibco), 1 mM sodium pyruvate (Gibco), 0.1 mM of each non-essential amino acids (Gibco), 0.1 mM 2-mercaptoethanol (Sigma), $10^4$ U mL$^{-1}$ penicillin/streptomycin (Gibco), and $10^3$ U mL$^{-1}$ mLIF (Gemini).

Cell lines were acquired from sources provided in the key resource table. All cell lines are routinely authenticated using ATCC STR service, and are tested for mycoplasma contamination on a monthly basis. All cell lines used for experiments tested negative.

## Western blot

Cells were treated with compounds as indicated and incubated for 24 h, or as indicated. Samples were run on 4–20%, AnyKD or 10% (in-vitro ubiqutination assay) SDS-PAGE gels (Bio-rad), and transferred to PVDF membranes using the iBlot 2.0 dry blotting system (Thermo Fisher Scientific). Membranes were blocked with LiCor blocking solution (LiCor), and incubated with primary antibodies overnight, followed by three washes in LiCor blocking solution and incubation with secondary antibodies for 1 h in the dark. After three final washes, the membranes were imaged on a LiCor fluorescent imaging station (LiCor). When Anti-mouse IgG, HRP Antibody was used, after three washes, the membranes were incubated with Amersham ECL Prime Western Blotting Detection Reagent for 1 min and subjected to imaging by Amersham Imager 600 (GE).

## Q5 mutagenesis and transient transfection

hsCRBN, hsSALL4, mmSALL4, and drSALL4 were PCR amplified and cloned into a pNTM-Flag based vector. Mutagenesis was performed using the Q5 site-directed mutagenesis kit (NEB, USA) with primers designed using the BaseChanger web server (http://nebasechanger.neb.com/).

Primer sets used for Q5 mutagenesis are:

hsSALL4 - S388N
Fwd 5′−3′: AAGTACTGTAaCAAGGTTTTTG
Rev 5′−3′: ACACTTGTGCTTGTAGAG
hsSALL4 – G416A
Fwd 5′−3′: TCTGTCTGTGcTCATCGCTTCAC
Rev 5′−3′: GCACACGAAGGGTCTCTC
hsSALL4 – G416N
Fwd 5′−3′: CTCTGTCTGTaaTCATCGCTTCACCAC
Rev 5′−3′: CACACGAAGGGTCTCTCT
hsSALL4 – G600A
Fwd 5′−3′: AAGATCTGTGcCCGAGCCTTTTC
Rev 5′−3′: ACACTGGAACGGTCTCTC
hsSALL4 – G600N
Fwd 5′−3′: TAAGATCTGTaaCCGAGCCTTTTCTAC
Rev 5′−3′: CACTGGAACGGTCTCTCC
Humanizing mmSALL4 – Y415F, P418S, I419V, L430F, Q435H
Fwd 5′−3′: AGGGCAATCTCAAGGTCCACTTtCAcCGACACCCTCAGGTGAAGGCAAACCCCC
Rev 5′−3′: TGGTGGTGAAGCGGTGACCACAGAcAGaGCACACGaAAGGTCTCTCTCCGGTGTG

For transient transfection, 0.2 million cells were seeded per well in a 12 well plate on day 1. On day 2, cells were transfected with 200–300 ng of plasmid (pNTM-Flag containing gene of interest) using 2 µL of lipofectamine 2000 transfection reagent (Invitrogen). On day 3, the desired concentration of IMiD was added to each well and cells were harvested after 24 h for western blot analysis using the protocol described above.

## Constructs and protein purification

His6DDB1ΔB (*Petzold et al., 2016*), His6-3C-SpyhsCRBN, His6-3C-SpyhsCRBN$^{V388I}$, Strep-BirAhsSALL4$_{590-618}$ (ZnF4), Strep-BirAhsSALL4$^{Q595H}_{590-618}$ (ZnF4), Strep-BirAhsSALL4$_{378-438}$ (ZnF1-2), Strep-BirAhsSALL4$_{402-436}$ (ZnF2), Strep-BirAmmSALL4$_{593-627}$ (ZnF4), Strep-BirAdrSALL4$_{583-617}$ (ZnF2) were subcloned into pAC-derived vectors or BigBac vector for HishsDDB1$_{1-1140}$-HishsCUL4A$_{38-759}$-HhismmRBX1$_{12-108}$ (CRL4$^{CRBN}$). Mutant Strep-BirAhsSALL4$_{378-438}$ (ZnF1-2) and Strep-BirAhsSALL4$_{402-436}$ (ZnF2) constructs were derived from these constructs using Q5 mutagenesis (NEB, USA). Recombinant proteins expressed in *Trichoplusia ni* High Five insect cells (Thermo Fisher Scientific) using the baculovirus expression system (Invitrogen). For purification of DDB1ΔB-CRBN$_{SpyBodipyFL}$ or CRL4$^{CRBN}$, cells were resuspended in buffer containing 50 mM tris(hydroxymethyl)aminomethane hydrochloride (Tris-HCl) pH 8.0, 200 mM NaCl, 1 mM tris(2-carboxyethyl)phosphine (TCEP), 1 mM phenylmethylsulfonyl fluoride (PMSF), 1× protease inhibitor cocktail (Sigma) and lyzed by sonication. Cells expressing variations of Strep-BirA-SALL4 were lyzed in the presence of 50 mM Tris-HCl pH 8.0, 500 mM NaCl, 1 mM TCEP, 1 mM PMSF, and 1× protease inhibitor cocktail (Sigma). Following ultracentrifugation, the soluble fraction was passed over appropriate affinity resin Ni Sepharose 6 Fast Flow affinity resin (GE Healthcare) or Strep-Tactin Sepharose XT (IBA), and eluted with 50 mM Tris-HCl pH 8.0, 200 mM NaCl, 1 mM TCEP, 100 mM imidazole (Fischer Chemical) for His$_6$-tagged proteins or 50 mM Tris-HCl pH 8.0, 500 mM NaCl, 1 mM TCEP, 50 mM D-biotin (IBA) for Strep-tagged proteins. Affinity-purified proteins were either further purified via ion exchange chromatography (Poros 50HQ) and subjected to size exclusion chromatography (SEC200 HiLoad 16/60, GE) (His6DDB1ΔB-His6-3C-SpyCRBN or CRL4$^{CRBN}$) or biotinylated overnight, concentrated, and directly loaded on the size exclusion chromatography (ENRich SEC70 10/300, Bio-rad) in 50 mM HEPES pH 7.4, 200 mM NaCl, and 1 mM TCEP. Biotinylation of Strep-BirASALL4 constructs was performed as previously described (*Cavadini et al., 2016*).

The protein-containing fractions were concentrated using ultrafiltration (Millipore), flash frozen in liquid nitrogen, and stored at −80°C or directly covalently labeled with BODIPY-FL-SpyCatcher$_{S50C}$ as described below.

## Spycatcher S50C mutant

Spycatcher (Zakeri et al., 2012) containing a Ser50Cys mutation was obtained as a synthetic dsDNA fragment from IDT (Integrated DNA technologies) and subcloned as a GST-TEV fusion protein in a pET-Duet-derived vector. Spycatcher S50C was expressed in BL21 DE3 and cells were lyzed in the presence of 50 mM Tris-HCl pH 8.0, 200 mM NaCl, 1 mM TCEP, and 1 mM PMSF. Following ultra-centrifugation, the soluble fraction was passed over glutathione sepharose 4B (GE Healthcare) and eluted with wash buffer (50 mM Tris-HCl pH 8.0, 200 mM NaCl, 1 mM TCEP) supplemented with 10 mM glutathione (Fischer BioReagents). The affinity-purified protein was TEV cleaved, subjected to size exclusion chromatography, concentrated, and flash frozen in liquid nitrogen.

## In vitro ubiquitination assays

In vitro ubiquitination was performed by mixing biotinylated SALL4 ZnF1-2 at 0.6 µM, and CRL4$^{CRBN}$ at 80 nM with a reaction mixture containing IMiDs at indicated concentrations or a DMSO control, E1 (UBA1, Boston Biochem) at 30 nM, E2 (UbcH5c, Boston Biochem and UBE2G1) at 1.0 µM each, ubiquitin (Ubiquitin, Boston Biochem) at 23 µM. Reactions were carried out in 50 mM Tris pH 7.5, 30 mM NaCl, 5 mM $MgCl_2$, 0.2 mM $CaCl_2$, 2.5 mM ATP, 1mM DTT, 0.1% Triton X-100 and 2.0 mg $mL^{-1}$ BSA, incubated for 60 min at 30°C and analyzed by western blot using rabbit anti-Strep-Tag II antibody at 1:10,000 (ab76949, Abcam) as described above.

## Lentiviral infection of mES cells

TC1 mES cells were transduced with a pCDH-MSCV-based lentiviral vector expressing hsCRBN, GFP, and the puromycin resistance gene. Infection was performed after 24 h in culture in a six-well 0.2% gelatin-coated plate using standard infection protocol in the presence of 2 µg $mL^{-1}$ polybrene (hexadimethrine bromide, Sigma). 72 h after transduction the cells were subjected to two rounds of puromycin selection (5 µg $mL^{-1}$) to form mES cells stably expressing hsCRBN, which were confirmed to be >90% GFP-positive under fluorescent microscope.

## Labeling of Spycatcher with Bodipy-FL-maleimide

Purified Spycatcher$_{S50C}$ protein was incubated with DTT (8 mM) at 4°C for 1 h. DTT was removed using a ENRich SEC650 10/300 (Bio-rad) size exclusion column in a buffer containing 50 mM Tris pH 7.5 and 150 mM NaCl, 0.1mM TCEP. Bodipy-FL-maleimide (Thermo Fisher Scientific) was dissolved in 100% DMSO and mixed with Spycatcher$_{S50C}$ to achieve 2.5 molar excess of Bodipy-FL-maleimide. SpyCatcher$_{S50C}$ labeling was carried out at room temperature (RT) for 3 h and stored overnight at 4°C. Labeled Spycatcher$_{S50C}$ was purified on an ENRich SEC650 10/300 (Bio-rad) size exclusion column in 50 mM Tris pH 7.5, 150 mM NaCl, 0.25 mM TCEP, and 10% (v/v) glycerol, concentrated by ultrafiltration (Millipore), flash frozen (~40 µM) in liquid nitrogen, and stored at −80°C.

## Bodipy-FL-Spycatcher labeling of CRBN-DDB1ΔB

Purified $_{His6}$DDB1ΔB-$_{His6-3C-Spy}$CRBN constructs (WT and V388I) were incubated overnight at 4°C with Bodipy-FL-maleimide-labeled SpyCatcher$_{S50C}$ protein at stoichiometric ratio. The protein was concentrated and loaded on the ENrich SEC 650 10/300 (Bio-rad) size exclusion column, and the fluorescence was monitored with absorption at 280 and 490 nm. Protein peak corresponding to the labeled protein was pooled, concentrated by ultrafiltration (Millipore), flash frozen in liquid nitrogen, and stored at −80°C.

## Time-resolved fluorescence resonance energy transfer (TR-FRET)

Compounds in binding assays were dispensed into a 384-well microplate (Corning, 4514) using the D300e Digital Dispenser (HP) normalized to 1% DMSO and containing 100 nM biotinylated strep-avi-SALL4 (WT or mutant, see Figure legends), 1 µM His$_6$-DDB1ΔB-His$_6$-CRBN$_{Bodipy-Spycatcher}$, and 4 nM terbium-coupled streptavidin (Invitrogen) in a buffer containing 50 mM Tris pH 7.5, 100 mM NaCl, 1mM TCEP, and 0.1% Pluronic F-68 solution (Sigma). Before TR-FRET measurements were conducted, the reactions were incubated for 15 min at RT. After excitation of terbium fluorescence at 337 nm, emission at 490 nm (terbium) and 520 nm (Bodipy) were recorded with a 70 µs delay over 600 µs to reduce background fluorescence, and the reaction was followed over 30× 200 s cycles of each data point using a PHERAstar FS microplate reader (BMG Labtech). The TR-FRET signal of each

data point was extracted by calculating the 520/490 nm ratios. Data from three independent measurements (n=3), each calculated as an average of five technical replicates per well per experiment, was plotted and the half-maximal effective concentrations $EC_{50}$ values calculated using variable slope equation in GraphPad Prism 7. Apparent affinities were determined by titrating Bodipy-FL-labelled DDB1ΔB-CRBN to biotinylated strep-avi-SALL4 (constructs as indicated) at 100 nM, and terbium-streptavidin at 4 nM. The resulting data were fitted as described previously (*Petzold et al., 2016*).

## Quantitative RT-PCR analysis

H9 hES cells treated with 10 μM thalidomide or DMSO for 24 h were subjected to gene expression analysis. RNA was isolated using the RNeasy Plus mini kit (Qiagen) and cDNA created by reverse transcription using ProtoScript II reverse transcriptase (NEB) following the manufacturer's instructions. The following primer sets from IDT were used with SYBR Green Master Mix (Applied Biosystems) to probe both GAPDH and total SALL4 levels:

SALL4total – F: GGTCCTCGAGCAGATCTTGT
SALL4total – R: GGCATCCAGAGACAGACCTT
GAPDH – F: GAAGGTGAAGGTCGGAGTC
GAPDH – R: GAAGATGGTGATGGGATTTC

Analysis was performed on a CFX Connect Real-Time PCR System (Bio-Rad) in a white 96-well PCR plate. Relative expression levels were calculated using the $\Delta\Delta C_T$ method.

## Sample preparation TMT LC-MS3 mass spectrometry

H9 hESC, Kelly, SK-N-DZ, and MM1s cells were treated with DMSO, 1 μM pomalidomide, 5 μM lenalidomid,e or 10 μM thalidomide in biological triplicates (DMSO) or biological duplicates (pomalidomide, lenalidomide, thalidomide) for 5 h, and cells were harvested by centrifugation. Lysis buffer (8 M urea, 50 mM NaCl, 50 mM 4-(2hydroxyethyl)-1-piperazineethanesulfonic acid (EPPS) pH 8.5, 1× Roche protease inhibitor, and 1× Roche PhosphoStop) was added to the cell pellets and cells were homogenized by 20 passes through a 21 gauge (1.25 in. long) needle to achieve a cell lysate with a protein concentration between 0.5 and 4 mg mL$^{-1}$. The homogenized sample was clarified by centrifugation at 20,000 × g for 10 min at 4°C. A micro-BCA assay (Pierce) was used to determine the final protein concentration in the cell lysate. 200 μg protein for each sample were reduced and alkylated as previously described (*An et al., 2017*). Proteins were precipitated using methanol/chloroform. In brief, four volumes of methanol were added to the cell lysate, followed by one volume of chloroform, and finally three volumes of water. The mixture was vortexed and centrifuged at 14,000 × g for 5 min to separate the chloroform phase from the aqueous phase. The precipitated protein was washed with three volumes of methanol, centrifuged at 14,000 × g for 5 min, and the resulting washed precipitated protein was allowed to air dry. Precipitated protein was resuspended in 4 M urea, 50 mM HEPES pH 7.4, followed by dilution to 1 M urea with the addition of 200 mM EPPS pH 8 for digestion with LysC (1:50; enzyme:protein) for 12 h at room temperature. The LysC digestion was diluted to 0.5 M urea, 200 mM EPPS pH 8, and then digested with trypsin (1:50; enzyme:protein) for 6 h at 37°C. Tandem mass tag (TMT) reagents (Thermo Fisher Scientific) were dissolved in anhydrous acetonitrile (ACN) according to manufacturer's instructions. Anhydrous ACN was added to each peptide sample to a final concentration of 30% v/v, and labeling was induced with the addition of TMT reagent to each sample at a ratio of 1:4 peptide:TMT label. The 10-plex labeling reactions were performed for 1.5 h at room temperature and the reaction quenched by the addition of 0.3% hydroxylamine for 15 min at room temperature. The sample channels were combined in a 1:1:1:1:1:1:1:1:1:1 ratio, desalted using C$_{18}$ solid phase extraction cartridges (Waters) and analyzed by LC-MS for channel ratio comparison. Samples were then combined using the adjusted volumes determined in the channel ratio analysis and dried down in a speed vacuum. The combined sample was then resuspended in 1% formic acid, and acidified (pH 2−3) before being subjected to desalting with C18 SPE (Sep-Pak, Waters). Samples were then offline fractionated into 96 fractions by high pH reverse-phase HPLC (Agilent LC1260) through an aeris peptide xb-c18 column (phenomenex) with mobile phase A containing 5% acetonitrile and 10 mM NH$_4$HCO$_3$ in LC-MS grade H$_2$O, and mobile phase B containing 90% acetonitrile and 10 mM NH$_4$HCO$_3$ in LC-MS grade H$_2$O (both pH 8.0). The

96 resulting fractions were then pooled in a non-continuous manner into 24 fractions or 48 fractions and every fraction was used for subsequent mass spectrometry analysis.

Data were collected using an Orbitrap Fusion Lumos mass spectrometer (Thermo Fisher Scientific, San Jose, CA, USA) coupled with a Proxeon EASY-nLC 1200 LC pump (Thermo Fisher Scientific). Peptides were separated on a 50 cm and 75 µm inner diameter Easyspray column (ES803, Thermo Fisher Scientific). Peptides were separated using a 3 h gradient of 6–27% acetonitrile in 1.0% formic acid with a flow rate of 300 nL/min.

Each analysis used an MS3-based TMT method as described previously (*McAlister et al., 2014*). The data were acquired using a mass range of *m/z* 350–1350, resolution 120,000, AGC target $1 \times 10^6$, maximum injection time 100 ms, dynamic exclusion of 90 s for the peptide measurements in the Orbitrap. Data-dependent MS2 spectra were acquired in the ion trap with a normalized collision energy (NCE) set at 35%, AGC target set to $1.8 \times 10^4$, and a maximum injection time of 120 ms. MS3 scans were acquired in the Orbitrap with a HCD collision energy set to 55%, AGC target set to $1.5 \times 10^5$, maximum injection time of 150 ms, resolution at 50,000, and with a maximum synchronous precursor selection (SPS) precursors set to 10.

### LC-MS data analysis

Proteome Discoverer 2.2 (Thermo Fisher) was used for RAW file processing and controlling peptide and protein level false discovery rates, assembling proteins from peptides, and protein quantification from peptides. MS/MS spectra were searched against a Uniprot human database (September 2016) with both the forward and reverse sequences. Database search criteria are as follows: tryptic with two missed cleavages, a precursor mass tolerance of 20 ppm, fragment ion mass tolerance of 0.6 Da, static alkylation of cysteine (57.02146 Da), static TMT labeling of lysine residues and N-termini of peptides (229.16293 Da), and variable oxidation of methionine (15.99491 Da). TMT reporter ion intensities were measured using a 0.003 Da window around the theoretical *m/z* for each reporter ion in the MS3 scan. Peptide spectral matches with poor-quality MS3 spectra were excluded from quantitation (summed signal-to-noise across 10 channels > 200 and precursor isolation specificity < 0.5). Reporter ion intensities were normalized and scaled using in-house scripts and the R framework (*R Core Team, 2013*). Statistical analysis was carried out using the limma package within the R framework (*Ritchie et al., 2015*).

### CRISPR/Cas9-mediated genome editing

For the generation of HEK293T [CRBN-/-] and Kelly[CRBN-/-] cells, HEK293T or Kelly cells were transfected with 4 µg of spCas9-sgRNA-mCherry using lipofectamine 2000. 48 h post transfection, pools of mCherry-expressing cells were obtained by fluorescence assisted cell sorting (FACS). Two independent pools were sorted to avoid clonal effects and artifacts specific to a single pool. For SALL4 antibody validation, HEK293T or Kelly cells were transfected with 4 µg of spCas9-sgRNA-mCherry using lipofectamine 2000. Protein levels were assessed by western blot 48 h post-transfection.

Guide RNA sequences used:
CRBN: TGCGGGTAAACAGACATGGC
SALL4-1: CCTCCTCCGAGTTGATGTGC
SALL4-2: ACCCCAGCACATCAACTCGG
SALL4-3: CCAGCACATCAACTCGGAGG

## Acknowledgements

We thank Michael J Eck, Stephen C Blacklow, and Nathanael S Gray for critical feedback on the manuscript. We thank Richard I Gregory, J Wade Harper, Nathanael S Gray, and Nicolas H Thomä for providing reagents. Financial support for this work was provided by NIH grant NCI R01CA214608 (grant to ESF), The Novartis/Dana-Farber Drug Discovery Program (grant to ESF), the Friends of Dana Farber (grant to ESF), and the Linde Family Foundation (start-up funds to ESF). Eric S Fischer is a Damon Runyon-Rachleff Innovator supported in part by the Damon Runyon Cancer Research Foundation (DRR-50–18).

## Additional information

### Competing interests

Eric S Fischer: is a member of the scientific advisory board of C4 Therapeutics. The other authors declare that no competing interests exist.

### Funding

| Funder | Grant reference number | Author |
|---|---|---|
| National Cancer Institute | R01CA214608 | Katherine A Donovan<br>Radoslaw P Nowak<br>Eric S Fischer |
| Novartis | | Katherine A Donovan<br>Bethany C Berry<br>Eric S Fischer |
| Damon Runyon Cancer Research Foundation | DRR-50-18 | Eric S Fischer |
| Linde Family Foundation | | Eric S Fischer |

The funders had no role in study design, data collection and interpretation, or the decision to submit the work for publication.

### Author contributions

Katherine A Donovan, Conceptualization, Data curation, Formal analysis, Investigation, Visualization, Methodology, Writing—original draft, Project administration, Writing—review and editing; Jian An, Formal analysis, Investigation, Visualization, Writing—review and editing; Radosław P Nowak, Data curation, Formal analysis, Investigation, Visualization, Methodology, Writing—review and editing; Jingting C Yuan, Emma C Fink, Bethany C Berry, Investigation; Benjamin L Ebert, Supervision, Writing—review and editing; Eric S Fischer, Conceptualization, Formal analysis, Supervision, Funding acquisition, Methodology, Writing—original draft, Project administration, Writing—review and editing

### Author ORCIDs

Emma C Fink (iD) http://orcid.org/0000-0002-6589-8558
Eric S Fischer (iD) http://orcid.org/0000-0001-7337-6306

### Decision letter and Author response

Decision letter https://doi.org/10.7554/eLife.38430.044
Author response https://doi.org/10.7554/eLife.38430.045

## Additional files

### Supplementary files

• Transparent reporting form
DOI: https://doi.org/10.7554/eLife.38430.032

### Data availability

All mass spectrometry raw data is deposited and made available via the PRIDE archive under accessions: PXD010416, PXD010417, PXD010418, PXD010420, PDX010428. Source files have been provided for all figures.

The following datasets were generated:

| Author(s) | Year | Dataset title | Dataset URL | Database, license, and accessibility information |
|---|---|---|---|---|
| Donovan KA, | 2018 | Thalidomide promotes degradation | https://www.ebi.ac.uk/ | Publicly available at |

| | | | | |
|---|---|---|---|---|
| Fischer ES | | of SALL4, a transcription factor implicated in Duane Radial Ray Syndrome, part 1 | pride/archive/projects/PXD010416 | EBI PRIDE (accession no: PXD010416) |
| Donovan KA, Fischer ES | 2018 | Thalidomide promotes degradation of SALL4, a transcription factor implicated in Duane Radial Ray Syndrome, part 2 | https://www.ebi.ac.uk/pride/archive/projects/PXD010417 | Publicly available at EBI PRIDE (accession no: PXD010417) |
| Donovan KA, Fischer ES | 2018 | Thalidomide promotes degradation of SALL4, a transcription factor implicated in Duane Radial Ray Syndrome, part 3 | https://www.ebi.ac.uk/pride/archive/projects/PXD010418 | Publicly available at EBI PRIDE (accession no: PXD010418) |
| Donovan KA, Fischer ES | 2018 | Thalidomide promotes degradation of SALL4, a transcription factor implicated in Duane Radial Ray Syndrome, part 5 | https://www.ebi.ac.uk/pride/archive/projects/PXD010420 | Publicly available at EBI PRIDE (accession no: PXD010420) |
| Donovan KA, Fischer ES | 2018 | Thalidomide promotes degradation of SALL4, a transcription factor implicated in Duane Radial Ray Syndrome, part 6 | https://www.ebi.ac.uk/pride/archive/projects/PXD010428 | Publicly available at EBI PRIDE (accession no: PXD010428) |

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
