## [Decision Letter]

Thank you for submitting your article "Thalidomide promotes degradation of SALL4, a transcription factor implicated in Duane Radial Ray Syndrome" for consideration by *eLife*. Your article has been reviewed by three peer reviewers, one of whom is a member of our Board of Reviewing Editors, and the evaluation has been overseen Didier Stainier as the Senior Editor. The reviewers have opted to remain anonymous.

The reviewers have discussed the reviews with one another and the Reviewing Editor has drafted this decision to help you prepare a revised submission.

Summary:

Thalidomide caused one of the most dramatic, yet almost entirely preventable, catastrophes in human biomedical history. Decades after it had been understood that thalidomide induced severe birth defects, such as phocomelia, thalidomide and related IMiDs are being used as powerful agents in treating multiple myeloma and myodysplastic syndrome. While the mechanism of action of IMiDs in cancer therapy, i.e. induction of degradation of two Zn-finger transcription factors or CK1, respectively, are beginning to be understood, the cause for thalidomide-induced birth defects remains unclear.

In this study, Fischer, Ebert, and colleagues used a proteomic approach to search for proteins that are being degraded in the presence of IMiDs in human embryonic stem cells (hESCs), as a model for human developmental disease. Notably, they identify the transcription factor SALL4 as a key new target of IMiDs that is degraded in the presence of thalidomide, lenalidomide, or pomalidomide. In addition, the authors identify a set of additional Zn-finger transcription factors as targets of more potent IMiDs, the degradation of which was dependent on the CRL4^CRBN^ machinery.

The findings of this study are important and deserve publication in *eLife*. Counter to what had been published before, they indicate that thalidomide also acts during development by inducing neo-substrate degradation, rather than by inhibiting the CRL4 machinery. This will spur research aimed at identifying the true developmental substrate of CRBN, a target which may be controlled by a small molecule related to IMiDs. Further, this work explains the previous species specificity of the IMiDs, a key issue that had contributed to the historic tragedy as mouse experiments did not reveal any noticeable toxicity (i.e. mouse SALL4 is not getting degraded by IMiDs). While this work does not prove that SALL4 is the key target, the genetics of human developmental diseases caused by heterozygous deletion of SALL4 provides a very strong argument for SALL4 being at least a strong contributor to IMiD-based birth defects.

While most experiments were performed at a high level, a few issues need to be carefully addressed before this manuscript can be officially accepted.

Required revisions:

1) Figure 3H and 3I are nice, but below the required quality for *eLife*. These need to be repeated, and Figure 3I needs the positive control on the same blot.

2) Figure 3J is also of fairly poor quality – it is also quite misleading, as it appears that the authors have used a no-lysine ubiquitin for these assays (which should be stated more directly in the legend or text). I would strongly recommend to repeat this assay with wildtype ubiquitin and enrich for ubiquitylated species (if they can't be detected directly) through TUBEs. This is a central figure for the paper, and it needs to be more convincing.

3) A critical experiment is missing for the argument of the authors that SALL4 represents a key substrate that might explain the teratogenic effects of thalidomide. They claim that both mutations in CRBN and the murine SALL4 degron prevent IMiD-dependent degradation. While one cannot expect to make a mouse that expresses both humanized CRBN and SALL4, it would be interesting to test whether mouse SALL4 with a human ZnF2 is degraded in a thalidomide-dependent manner in Kelly cells. This would provide stronger evidence for the authors' arguments, and given the genetics of DRRS, it would provide a very strong case for SALL4 being the key target of thalidomide in causing birth defects.

---

## [Author Response]

Required revisions:1) Figure 3H and 3I are nice, but below the required quality for eLife. These need to be repeated, and Figure 3I needs the positive control on the same blot.

Figures 3H and 3I (now 3G and 3H) have been repeated, and blotting conditions optimized using fluorescence detection similar to all other western blots in this study. Both new Figures 3G and 3H each now contain the SALL4^wt^ control on the same blot.

2) Figure 3J is also of fairly poor quality – it is also quite misleading, as it appears that the authors have used a no-lysine ubiquitin for these assays (which should be stated more directly in the legend or text). I would strongly recommend to repeat this assay with wildtype ubiquitin and enrich for ubiquitylated species (if they can't be detected directly) through TUBEs. This is a central figure for the paper, and it needs to be more convincing.

We thank the reviewers for their suggestion and have repeated the experiment using wt ubiquitin. Figure 3J has been replaced with the new experimental data using the wild type ubiquitin to enable visualization of polyubiquitin chains on SALL4.

3) A critical experiment is missing for the argument of the authors that SALL4 represents a key substrate that might explain the teratogenic effects of thalidomide. They claim that both mutations in CRBN and the murine SALL4 degron prevent IMiD-dependent degradation. While one cannot expect to make a mouse that expresses both humanized CRBN and SALL4, it would be interesting to test whether mouse SALL4 with a human ZnF2 is degraded in a thalidomide-dependent manner in Kelly cells. This would provide stronger evidence for the authors' arguments, and given the genetics of DRRS, it would provide a very strong case for SALL4 being the key target of thalidomide in causing birth defects.

We thank the reviewers for this insight. We have created a mmSALL4 construct containing the human ZnF2 (point mutation of five amino acids in the ZnF2 of mouse SALL4) and now show that the ‘humanized’ mouse SALL4 is degraded in a thalidomide-dependent manner in human Kelly cell (new Figure 4G).